# Choroidal Thickness Increase after Subliminal Transscleral Cyclophotocoagulation

**DOI:** 10.3390/diagnostics12071513

**Published:** 2022-06-21

**Authors:** Florian Baltă, Valentin Dinu, Mihail Zemba, George Baltă, Andreea Diana Barac, Speranța Schmitzer, Christiana Diana Maria Dragosloveanu, Ramona Ileana Barac

**Affiliations:** 1Faculty of Medicine, “Carol Davila” University of Medicine and Pharmacy, 050474 Bucharest, Romania; florian.balta@umfcd.ro (F.B.); mhlzmb@yahoo.com (M.Z.); andreea.barac@stud.umfcd.ro (A.D.B.); speranta.sch@gmail.com (S.S.); christianacelea@gmail.com (C.D.M.D.); ramona.barac@live.com (R.I.B.); 2Bucharest Emergency Eye Hospital, 030167 Bucharest, Romania; baltageorge@gmail.com; 3Ophthalmology Department, “Dr. Carol Davila” Central Military Emergency University Hospital, 010825 Bucharest, Romania

**Keywords:** subliminal transscleral cyclophotocoagulation, choroidal thickness, intraocular pressure

## Abstract

Background: The purpose of this study is to estimate the success rate of subliminal transscleral cyclophotocoagulation for refractory glaucoma and to determine the correlation between the decrease in intraocular pressure and the variation in choroidal thickness. Methods: A pre–post study was conducted over a period of 3 years, including 81 eyes from 67 patients with different types of drug-refractory glaucoma who underwent subliminal transscleral cyclophotocoagulation. The variables included best-corrected visual acuity, intraocular pressure and choroidal thickness. Results: We observed the following success rates (defined as IOP < 21 mmHg): 80% at 1 month (65 patients), 74% at 3 months (60 patients), 64% at 6 months (52 patients) and 50.6% at 1 year (41 patients). A strong correlation was noted between the decrease in intraocular pressure and the increase in the average choroidal thickness at 1 year (318.42 µm) compared to the average preoperative thickness (291.78 µm). A correlation of increased choroidal thickness at 1-month with the success rate of the procedure was also observed. Conclusions: We observed a statistically significant correlation between the success rate, decrease in intraocular pressure and choroidal thickness. The correlation of increased choroidal thickness at 1-month with the success rate of the procedure could be used clinically as a predictive factor for the final outcome of patients. Further experimental research is warranted to determine whether the increase in choroidal thickness after subliminal transscleral cyclophotocoagulation is indeed evidence of increased uveoscleral drainage.

## 1. Introduction

Glaucoma is an ophthalmic pathology frequently encountered in current medical practice, with a high prevalence of patients with high intraocular pressure (IOP) or glaucomatous progression despite maximum antiglaucoma drug treatment. Micropulsed (Iridex, Silicon Valley, CA, USA) or subliminal (Quantel Medical, Cournon-d’Auvergne, France) transscleral cyclophotocoagulation is a new, noninvasive and non-incisional technique that bridges the gap between medical and surgical antiglaucomatous therapy.

There are at least three important mechanisms for lowering IOP using this technique: inflammation, decreased production of aqueous humor and increased uveoscleral elimination. The uveoscleral pathway represents 10% to 20% of the aqueous humor drainage pathway, and it is independent of intraocular pressure. State-of-the-art enhanced depth imaging optical coherence tomography (EDI-OCT) and swept source optical coherence tomography (SS-OCT) have made it possible to clearly visualize the entire choroidal structure and to measure its thickness [1,2,3].

Using SS-OCT technology, we sought to estimate the success rate of subliminal transscleral cyclophotocoagulation for glaucoma and to determine the correlation between the decrease in intraocular pressure and the variation in choroidal thickness.

## 2. Materials and Methods

We conducted a prospective non-randomized pre–post study over a period of 3 years (2018–2021) including eyes with various types of drug-refractory glaucoma that were not amenable to surgery. Patients with significant cardiovascular issues and congenital/juvenile forms of glaucoma were excluded. This study was approved by the Ethics Committee of Ponderas Academic Hospital, Bucharest, Romania and was conducted in accordance with all principles of the World Medical Association’s Declaration of Helsinki. Written informed consent was obtained from all patients.

The study included 81 eyes from 67 patients with a one year follow-up. In each case, we performed subliminal transscleral cyclophotocoagulation (Quantel laser, SubCyclo laser procedure, subliminal transscleral cyclophotocoagulation, Lumibird, Lannion, France). The following data were obtained from each patient: best-corrected visual acuity (BCVA), IOP and choroidal thickness as measured by SS-OCT (TOPCON, Tokyo, Japan). IOP was measured using the same Goldmann tonometer. Choroidal thickness measurements were performed at the level of the foveal depression. Depending on the OCT appearance, the thickness of the choroid was measured from Bruch’s membrane to the choroid-sclera interface, to the suprachoroidal space or to the external limit of the large choroidal vessels (Figure 1).

We performed these measurements in all 81 patients preoperatively and postoperatively at 1 month, 3 months, 6 months and 1 year. Numerous studies [1,2,4] have described variations in choroidal thickness depending on the circadian rhythm and vascular factors, which is why all patients included in the study were within normal cardiovascular parameters and were examined in the morning to eliminate errors related to these variations.

The cyclophotocoagulation procedure was performed with parabulbar anesthesia (lidocaine 4% and ropivacaine) and by moving the probe 360° at approximately 3 mm from the sclerocorneal limbus, avoiding the 3 and 9 o’clock meridians. The total exposure time was 240 s (120 s × 2). The default characteristics of the laser were as follows: duty cycle of 31.3% and power of 2000 mW.

After cyclophotocoagulation, all patients received topical nonsteroidal anti-inflammatory drops and continued previous antiglaucomatous therapy until the first check-up at one month. Statistical analyses were conducted using SPSS 23.0 (IBM Corp., Armonk, NY, USA); statistical significance was set at *p* < 0.05. IOP, and choroidal thickness at different time points of follow-up are described using the mean and standard deviation. To assess the correlations between the difference in IOP and the difference in choroidal thickness, the Pearson correlation coefficient was used.

## 3. Results

For this study, 81 eyes from 67 patients with glaucoma refractory to maximum drug treatment were selected. All patients exhibited well-compensated cardiovascular status. The distribution by sex was approximately equal (49% women vs. 51% men), and there were no significant differences in the distributions of glaucoma by sex (chi-square = 2.33; *p* = 0.675). No significant sex-related differences in BCVA were observed at any given time point before or after the SubCyclo laser procedure. A large difference was observed in the IOP before and after treatment, even at one month (Figure 2). At one year, the average IOP was half the average preoperative IOP.

The distribution of cases according to type of glaucoma was assessed. The results were as follows: 35 patients (43.2%) had primary open-angle glaucoma, 21 patients (25.9%) had secondary neovascular glaucoma, 17 patients (21.0%) had glaucoma secondary to vitreoretinal surgery, seven patients (8.6%) had secondary inflammatory glaucoma, and only one patient (1.2%) had post-traumatic glaucoma. (Table 1).

We defined the success rate at 1 month, 3 months, 6 months and 1 year as the percentage of patients with an IOP measurement below 21 mmHg. The success rate was 80% at 1 month (65 patients), 74% at 3 months (60 patients), 64% at 6 months (52 patients) and 50.6% at 1 year (41 patients). Regarding choroidal thickness, an increase from the average preoperative value (291.78 µm) was observed at 1 year (318.42 µm). Even though the increase observed at 1 month postoperatively did not remain at the same values until the last assessment, when compared to the preoperative baseline value, an increase in choroidal thickness was still observed.

A correlation was observed between the success rate and an increase in choroidal thickness. Overall, 88% of the patients responsive at 3 months demonstrated an increase in choroidal thickness at 1 month. The same percentage was observed for the patients responsive at 6 months. On the other hand, only 57% of unresponsive patients showed an increase in choroidal thickness at 1 month. Moreover, 67% of patients with success at 3 months, but not maintaining success at 6 months, showed a decrease in choroidal thickness (Table 2).

The decrease in IOP due to SubCyclo is also supported by the decrease in the average daily number of antiglaucomatous drops used (3.69 preoperatively, 3.61 at 1 month, 3.35 at 3 months, 3.26 at 6 months and 2.7 at 1 year). With the same trend, the average daily number of acetazolamide tablets used (250 mg each) decreased gradually postoperatively (1.197 preoperatively, 1.197 at one month, 0.605 at 3 months, 0.45 at 6 months and 0.106 at 1 year).

## 4. Discussion

The aim of this study was to determine the correlation between the decrease in IOP and the variations in choroidal thickness in patients with initially uncompensated glaucoma, in whom the SubCyclo procedure was performed. The IOP results presented above emphasize the effectiveness of the procedure both in the short and long term. Similar results in all types of glaucoma indicate that this technique could be an effective solution for more patients. Given the inherent bias of this study towards recalcitrant glaucoma, the SubCyclo laser procedure allowed us to reach our therapeutic goal in cases where other procedures are deemed inadequate.

The significant increase in choroidal thickness after treatment supports the hypothesis that increased uveoscleral outflow may indeed be a key IOP-lowering mechanism of the SubCyclo procedure. The correlation of increased choroidal thickness at 1-month with the success rate of the procedure could be used clinically as a predictive factor for the final outcome of each case. The effectiveness of subliminal [5] or micropulsed transscleral cyclophotocoagulation [6,7,8,9,10] has been demonstrated in numerous studies. It is an undemanding intervention for both doctors and patients.

Thus far, there are several papers that describe an increase in choroidal thickness after trabeculoplasty [11,12]. We do not know exactly whether this is determined by the same pathophysiological mechanism as described in this article. The choroid is one of the eye structures that has been insufficiently studied thus far, although SS-OCT has been used extensively to gauge ocular anatomy and its response to therapy [3]. We previously performed and published several studies showing the correlation between choroid thickness and prostaglandin analogue administration and the performance of micropulsed transscleral cyclophotocoagulation [13].

The aim of our study was to demonstrate the correlation between increased choroidal thickness and decreased IOP after the SubCyclo procedure, confirming the mechanism of increased uveoscleral drainage of aqueous humor after SubCyclo. New imaging techniques, such as OCT angiography or adaptive optics, have allowed the imaging of retinal vessels and choroidal vessels [14]. We intend to continue this study to establish the success rate at 2 years and to conduct further studies using OCT angiography and adaptive optics.

In conclusion, transscleral cyclophotocoagulation is a procedure with rapid post-intervention recovery, a satisfactory success rate, minimal possible complications and the possibility of repeated treatments.

## Figures and Tables

**Figure 1 diagnostics-12-01513-f001:**
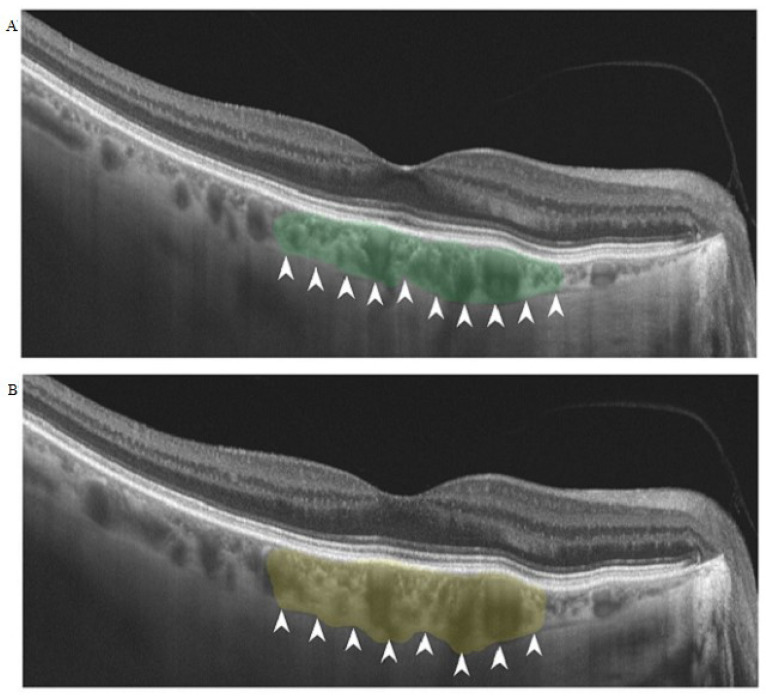
Choroidal thickness before and after the subliminal transscleral cyclophotocoagulation. The white arrows indicate the edge of the choroid. (**A**) Choroid thickness before treatment; choroidal subfoveolar thickness (green area) is 174 µm. (**B**) Choroid thickness after treatment; choroidal subfoveolar thickness (yellow area) is 205 µm.

**Figure 2 diagnostics-12-01513-f002:**
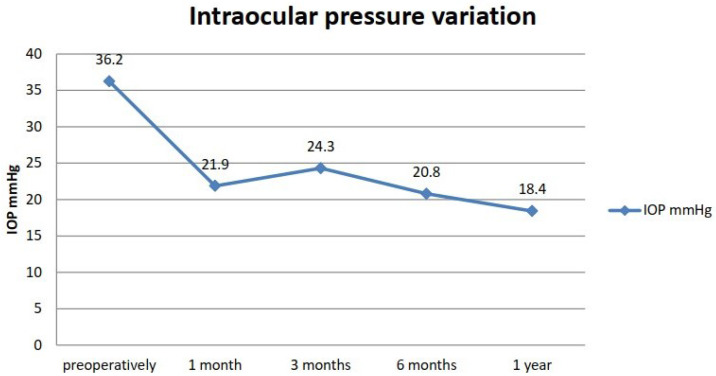
Intraocular pressure variation.

**Table 1 diagnostics-12-01513-t001:** Variation of the IOP and choroidal thickness depending on the etiology of glaucoma.

Moment	IOP/ Choroid	Primary Open-Angle Glaucoma	Post-Traumatic	Secondary Neovascular Glaucoma	Secondary to Vitreoretinal Surgery	Secondary Inflammatory Glaucoma	*p* Value
Preoperative	IOP	30.37 ± 7.93	24.00	42.76 ± 11.46	38.17 ± 9.58	43.14 ± 8.91	<0.001
	Choroid	302.08 ± 92.58	445.00	277.62 ± 79.11	226.56 ± 98.97	410.00 ± 119.74	<0.001
1 month	IOP	18.34 ± 7.97	13.00	25.52 ± 9.81	22.71 ± 10.27	26.86 ± 15.47	0.044
	Choroid	328.83 ± 102.34	596.00	336.19 ± 112.69	275.35 ± 151.05	450.71 ± 142.21	0.006
3 months	IOP	20.83 ± 8.56	13.00	28.81 ± 13.73	25.47 ± 13.36	26.86 ± 11.61	0.103
	Choroid	334 ± 03 ± 104.01	589.00	315.90 ± 108.75	267.29 ± 117.58	442.14 ± 167.88	0.003
6 months	IOP	19.46 ± 7.86	21.00	22.94 ± 8.57	20.75 ± 7.81	19.67 ± 11.23	0.741
	Choroid	325.27 ± 118.14	584.00	315.66 ± 110.52	291.08 ± 85.83	431.00 ± 156.97	0.089
12 months	IOP	16.67 ± 3.82	18.00	20.67 ± 7.38	19.62 ± 5.57	15.50 ± 2.12	0.254
	Choroid	311.24 ± 98.79	590.00	337.93 ± 140.05	247.37 ± 100.82	396.00 ± 291.33	0.082

**Table 2 diagnostics-12-01513-t002:** Correlation between the IOP, choroidal thickness and success rate in all cases.

Moment	IOP	Choroid	Success Rate		
	IOP (mmHg)	*p* Value	Choroid Thickness (µm)	*p* Value	
Baseline	36.24 ± 10.81	-	291.78 ± 103.54	-	-
1 month	21.79 ± 10.11	<0.001	333.35 ± 128.45	<0.001	65p (80%)
3 months	24.29 ± 11.71	0.002	327.81 ± 123.54	<0.001	60p (74%)
6 months	20.80 ± 8.08	0.114	323.35 ± 114.99	<0.001	52p (64%)
12 months	18.42 ± 5.61	0.661	318.43 ± 128.44	<0.001	41p (50,6%)
Trend *p* value */**	0.001 */ 0.009 **		0.216 */ 0.921 **		0.039 **

*p* * = *p* values of the trend including baseline values, *p* ** = *p* values of the trend disregarding baseline values (among post intervention values).

## Data Availability

The first author and the corresponding author have full access to all the data and materials in the study.

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
