# Peer review of "Choroidal Thickness Increase after Subliminal Transscleral Cyclophotocoagulation"

_diagnostics, 2022, doi:10.3390/diagnostics12071513_

Round 1

Reviewer 1 Report

The paper is well prepared and written. The purpose of this study was to estimate the success rate of subliminal  transscleral cyclophotocoagulation for refractory glaucoma and to determine the correlation between  the decrease in intraocular pressure and the variation in choroidal thickness. However, although the authors mentioned that they demonstrated the correlation between increased choroidal thickness and decreased IOP after the SubCyclo  procedure can be  confirmed by uveoscleral drainage of aqueous humor, they did not bring up the evidence. Thus, it is advised that the authors demonstrate more evidence for their speculation for this paper.

Author Response

Dear sir,

Thank you for your revision. The increased uveoscleral outflow is the mechanism believed to be related to the IOP reduction after an MP laser. We will add some citations to this effect. 

We will also improve the Materials and Methods section and do a general spellcheck of the paper.

Reviewer 2 Report

Florian Balta  et al present an interesting study that  CHOROIDAL THICKNESS INCREASE AFTER SUBLIMINAL TRANSSCLERAL CYCLOPHOTOCOAGULATION 

The  study results certainly suggest some degrees that observed a statistically significant correlation between success rate, decrease in 15 intraocular pressure, and choroidal thickness. The increase in choroidal thickness after subluminal 16 transscleral cyclophotocoagulation is evidence of increased uveoscleral drainage. 

Besides how the magnitude of these data add new findings compare to the current standard can not be determined based on this study 

The results are encouraging and further study is warranted. 

here some relevant points :

1. please add on the keywords this does not match with the manuscript 

2. The authors should  express why is relevant CHOROIDAL THICKNESS INCREASE AFTER SUBLIMINAL TRANSSCLERAL CYCLOPHOTOCOAGULATION 

What does it Change for the current standard of care  ?

3. The authors should explain why their findings make a different for ophthalmologist  around the world and for the readers of MDPI

4. The authors should explain the source of the information  and what were the criteria they used for adding to the paper  Were the assessors masked? What was the ICC between them in orden to analyzed the data ? I Was the randomization digitalized?  if not please clarified.

5.  Please add references and rephrase the sentence. English grammar should be applied. 

 “Moreover, if we consider that glaucoma was refractory at baseline, and we achieved a success rate of 80% at 1 month and 50.6% at 1 year—even with the reduction of medication—then the SubCyclo laser procedure allowed us to reach the therapeutic goal in almost all cases. The significant increase in choroidal thickness after treatment in all types of glaucoma”

6.   please add in the introduction that papers have been published showing the important of tool to diagnose CHOROIDAL THICKNESS  properly and paper published showing the important of theses and how this will help physician around the world to proper diagnosis , add one line in the introduction of this and also in the discussion section  These papers should be describe in this general considerations. 

Reference :

-Retina. 2019 Jan;39(1):44-51. doi: 10.1097/IAE.0000000000002196.

DEXAMETHASONE IMPLANT FOR DIABETIC MACULAR EDEMA IN NAIVE COMPARED WITH REFRACTORY EYES: The International Retina Group Real-Life 24-Month Multicenter Study. The IRGREL-DEX Study.

Iglicki M1, Busch C2, Zur D3,4, Okada M5, Mariussi M6, Chhablani JK7, Cebeci Z8, Fraser-Bell S9, Chaikitmongkol V10, Couturier A11, Giancipoli E12, Lupidi M13, Rodríguez-Valdés PJ14, Rehak M2, Fung AT15,16,17, Goldstein M3,4, Loewenstein A3,4,17.

English Grammar should be applied .This paper should be corrected by an english redactor 

Author Response

Dear sir,

thank you for your revision.

  1. Our Keywords are subliminal transscleral cyclophotocoagulation; choroidal thickness; intraocular pressure. We feel these are adequate.
  2. We agree we have not clarified the value of this correlation. We will expand on this topic.
  3. We agree we have not clarified the value of this correlation. We will expand on this topic.
  4. The eyes are not randomized, since this is not a randomized study. There is no control group, rather each eye serves as its own control from baseline to the end of the follow-up
  5. The sentence will be revised.
  6. Alright